# In Vitro Mechanism Assessment of Zearalenone Removal by Plant-Derived *Lactobacillus plantarum* BCC 47723

**DOI:** 10.3390/toxins13040286

**Published:** 2021-04-19

**Authors:** Saowalak Adunphatcharaphon, Awanwee Petchkongkaew, Wonnop Visessanguan

**Affiliations:** 1School of Food Science and Technology, Faculty of Science and Technology, Thammasat University, 99 Mhu 18, Pahonyothin Road, Khong Luang, Pathum Thani 12120, Thailand; s.adunphatcharaphon@hotmail.com (S.A.); awanwee@tu.ac.th (A.P.); 2National Center for Genetic Engineering and Biotechnology (BIOTEC), National Science and Technology Development Agency (NSTDA), 113 Thailand Science Park, Phahonyothin Road, Pathum Thani 12120, Thailand

**Keywords:** mycotoxin, zearalenone, mycotoxin adsorption, lactic acid bacteria, plant-derived lactic acid bacteria, *Lactobacillus plantarum*

## Abstract

Zearalenone (ZEA) is a harmful secondary fungal metabolite, produced primarily by plant pathogenic fungi mostly belonging to the genus *Fusarium*. It is involved in reproductive disorders in animals since its structure is similar to the estrogen hormone. This induces precocious pubertal changes, fertility problems, and hyper estrogenic disorders. The main objectives of this study were to evaluate the ZEA removal capacity of plant-derived lactic acid bacteria (LAB) and to investigate the possible components and mechanisms involved in the removal of ZEA by physically and chemically treated plant-derived LAB. The bacterial cells were characterized using scanning electron microscopy coupled with energy dispersive X-ray spectroscopy (SEM-EDS), Fourier transform infrared spectroscopy (FTIR), and the analysis of zeta potential, and hydrophobic index. Results revealed that 17 out of 33 plant-derived LAB exhibited ZEA removal from liquid medium. The percentage of removal ranged from 0.5–23% and *Lactobacillus plantarum* BCC 47723, isolated from wild spider flower pickle (Pag-sian-dorng), exhibited the highest removal. The alteration of proteins on *L. plantarum* BCC 47723 structure by Sodium dodecyl sulphate (SDS) treatment was positively affected on ZEA removal, whereas that of lipids on ZEA removal was negatively observed. Heat treatment influenced the higher ZEA adsorption. SEM images showed that the morphologies of modified bacterial cells were distinctly deformed and damaged when compared with untreated control. FTIR analysis indicated that the original functional groups, which included amide (C=O, C-N), carboxyl (C=O, C-O, O-H), methylene (C=C), and alcohol (O-H) groups, were not changed after ZEA adsorption. The zeta potential indicated that electrostatic interaction was not involved in the ZEA removal, while hydrophobicity was the main force to interact with ZEA. These findings can conclude that adsorption by hydrophobicity is the main mechanism for ZEA removal of plant-derived *L. plantarum* BCC 47723. The alteration of bacterial cell structure by heat treatment enhanced the efficiency of *L. plantarum* BCC 47723 for ZEA reduction. Its activity can be protected by the freeze-drying technique. Hence, plant-derived *L. plantarum* BCC 47723 can be considered as an organic adsorbent for ZEA reduction in food and feedstuff.

## 1. Introduction

Nowadays, agricultural foodstuffs and animal feed worldwide is highly threatened by mycotoxins. The consumption of a mycotoxin-contaminated diet causes disorder in human and animal health [1,2,3,4,5]. Zearalenone (ZEA) is one harmful mycotoxin produced by field *Fusarium* fungi in temperate and warm countries [6,7]. It has been extensively detected in cereal grains including maize, soybean, wheat, barley, beer, and feed [1,8,9,10,11,12,13]. Since the molecular structure of ZEA and its derivatives is similar to the estrogen hormone, they can competitively bind to the estrogen receptors, resulting in reproductive tract disorder [3]. Swine are the most sensitive animal affected on the farm [14]. Furthermore, ZEA has also been reported to be immunotoxic, hepatotoxic, hematotoxic, and nephrotoxic and can induce clinical signs such as swelling of the vulva, vaginal and rectal prolapses, and alterations within the uterus [5,15,16]. For these reasons, ZEA is considered as one of the significant mycotoxins which must be controlled in foodstuffs and animal feed [17,18].

Numerous physical and chemical strategies for ZEA elimination including extrusion, milling, washing, adsorbents, ozone (O_3_), and hydrogen peroxide (H_2_O_2_) treatment have been reported [19,20,21,22]. Nevertheless, these strategies have some limitations concerning losses of important nutrients and have high machinery and maintenance cost [5,13]. Therefore, the biological strategy is one of the promising techniques for ZEA reduction because of its advantages in efficiency, specificity, and environmental soundness [14,23]. There are two mechanisms, comprised of transformation and adsorption, involved in ZEA reduction by microorganisms [24,25,26,27,28]. The fungus *Gliocladium roseum* showed the capability to reduce ZEA by cleaving the lactone ring through lactonohydrolase enzyme [25,26,29] and the yeast *Trichosporon mycotoxinivorans* is able to transform ZEA to a non-toxic structure [27]. In the case of adsorption, several lactic acid bacteria (LAB) either originally isolated from human, or animal, or dairy products such as *Lactobacillus rhamnosus*, *Lactobacillus curvatus*, *Lactobacillus casei*, *Lactobacillus brevis*, *Lactobacillus plantarum*, and *Lactobacillus pentosus* exhibit ZEA removal from a liquid medium [24,28,30,31,32]. El-Nezami et al. [31] suggested that binding is the main mechanism of *Lactobacillus rhamnosus* for ZEA elimination. ZEA, in all likelihood, binds with carbohydrate and protein on the bacterial cell wall surface and hydrophobic interactions play a role in the ZEA binding mechanism. Although the mechanism of ZEA removal was indicated throughout this publication, some questions still remained such as the adsorption of other LAB species or the exact mechanism of ZEA adsorption. Therefore, to have an in-depth understanding of the ZEA removal by other LAB species, plant-derived LAB which can survive in a much harsher environment was assessed for ZEA removal. The characterization of the mechanism was investigated by the exploitation of scanning electron microscopy coupled with energy dispersive X-ray spectroscopy (SEM-EDS), Fourier transform infrared spectroscopy (FTIR), and the measurement of zeta potential, and hydrophobic index.

## 2. Results

### 2.1. Screening of Plant-Derived LAB for ZEA Removal

A total of 33 plant-derived LAB strains, isolated from Thai fermented vegetables, were assessed for ZEA removal at an optimum temperature of 30 °C for 1 h. There were 17 strains which were able to remove ZEA from buffer solution. The percentage of ZEA removal capacity by these strains is shown in Figure 1. Their activity ranged from 0.5–23%. *L. plantarum* BCC 47723, which was isolated from wild spider flower pickle (Pag-sian-dorng), exhibited the highest percentage of ZEA removal (23.3%), followed by *Lactobacillus namurensis* (21.4%), *Lactobacillus brevis* (18.0%), and *Pediococcus pentosaceus* (17.6%), respectively. The ZEA removal capacity by *L. plantarum* BCC 47723 was confirmed by UHPLC analysis. The result showed that its reduction was approximately 25% of ZEA and no degradation products were observed (data not shown). Therefore, *L. plantarum* BCC 47723 was chosen for further experiments.

### 2.2. Effect of Physical and Chemical Treatments

To investigate the effect of cell components on ZEA removal, *L. plantarum* BCC 47723 was treated with eight different treatments, which are shown in Figure 2. The viable cell of *L. plantarum* BCC 47723 was used as an untreated control. Three treatments, including SDS, lipase, and heat, significantly affected the ZEA removal of bacterial cells. As shown in Figure 2, SDS treatment exhibited the highest impact on ZEA removal (55%), followed by heat treatment (53%). They significantly enhanced the capacity of bacterial cells to remove ZEA (*p* < 0.05), whereas lipase treatments significantly decreased the ZEA reduction capability (*p* < 0.05). For other treatments, it was shown that those treatments slightly affected the ZEA removal by bacterial cells (Figure 2).

### 2.3. Characterization of ZEA Removal Mechanism by L. plantarum BCC 47723

To understand the mechanism of *L. plantarum* BCC 47723 on ZEA removal, both ZEA-exposed and -unexposed bacterial cells from the previous experiment were characterized. Scanning electron microscopy coupled with energy dispersive X-ray spectroscopy (SEM-EDS) was used to investigate the morphology and elementary composition, whereas Fourier transform infrared spectroscopy (FTIR) was applied to estimate the possible functional groups and adsorption sites which were involved in ZEA removal. Electrostatic force and hydrophobic interaction were assessed in order to investigate the adsorption mechanism of *L. plantarum* BCC 47723 through the measurement of zeta potential and surface hydrophobicity (H_0_).

SEM-EDS was used for the investigation of the morphology and elementary composition of *L. plantarum* BCC 47723 after treated with physical and chemical treatments. The alteration of bacterial cells in different treatments was observed, as shown in Figure 3. SEM photographs at a magnification of 20,000× showed that the morphologies of modified bacterial cells were distinctly deformed and damaged, when compared to the untreated control. The change of the bacterial cell surface after physical and chemical treatments certainly affected the ZEA removal capacity. The EDS analysis showed that C, O, N, P, K, and Na were the main elements on the bacterial cell surface and the carbon concentration was the highest, followed by O, N, P, K, and Na (Table 1). The physical and chemical treatments were not significantly influenced by the atomic concentration when compared to the untreated control.

The functional groups present on bacterial cell surface of *L. plantarum* BCC 47723 in eight treatments were identified using FTIR. In this analysis, spectra were examined in the range v = 4000–500 cm^−1^. The FTIR spectra of ZEA-exposed and -unexposed bacterial cells in all treatments are shown in Figure 4. The peak vibrations of ZEA-unexposed and -exposed bacterial cells were not different, except the peak vibration of the heat treatment (Figure 4). Apparently, ZEA removal by *L. plantarum* BCC 47723 did not completely lose its original structure when compared with the untreated control. The main compound structures of the bacterial cells remained after reaction with ZEA, which included amide (C=O, C-N), carboxyl (C=O, C-O, O-H), methylene (C=C), and alcohol (O-H) groups.

As shown in Figure 5, the zeta potentials of ZEA-unexposed bacterial cells ranged from −14.27 to −16.27 mV, lower than the values of ZEA-exposed bacterial cells (−13.13 to −14.80 mV). No significant difference (*p* ≥ 0.05) was observed in each treatment of ZEA-unexposed and -exposed bacterial cells. When comparing between ZEA-unexposed and -exposed bacterial cells, the results showed that the charge of SDS and heat-treated cells was significantly decreased (*p* ˂ 0.05) after cells interacted with ZEA. 

The surface hydrophobicity (H_0_) of ZEA-unexposed and -exposed bacterial cells of all treatment was determined and is shown in Figure 6. The data indicated that the physical and chemical treatments affected the H_0_ values. Treating the cells with urea, SDS, and heat significantly enhanced the H_0_ values of bacterial cells, whereas m-periodate, polymyxin B, pronase E, and lipase treatments were not significantly different when compared with the untreated control. Heat treatment affected the H_0_ of bacterial cells more than other treatments (*p* < 0.05). Additionally, the results also indicated that the H_0_ values in all treatments decreased after the cells were treated with ZEA. 

Pearson’s correlation coefficient (PCC) was used to assess the correlation among the atomic concentration, zeta potential, and surface hydrophobicity (H_0_) in ZEA removal by *L. plantarum* BCC 47723. As shown in Table 2, H_0_ was closely associated with the ZEA removal by bacterial cells (*p* < 0.05) while other factors were not. This indicated that hydrophobicity is importantly involved in ZEA removal. The ZEA removal was also related to the potassium value (K), at *p* < 0.01.

### 2.4. Shelf-Life of the Bacterial Cells on ZEA Removal

Lyophilization or freeze-drying was used to investigate the shelf-life of heat-inactivated cells for ZEA removal. The results showed that ZEA removal capability still remained more than 30% after being kept in a desiccator at room temperature for 90 days (Figure 7). 

## 3. Discussion

LAB have been extensively found in fermented foods (meat, vegetable, and milk), non-fermented foods, and in the intestinal and respiratory tracts of humans and animals [33,34]. The capability of LAB to remove mycotoxins has been widely reported, including aflatoxin B_1_ (AFB_1_) [24,35,36,37], zearalenone (ZEA) [30,31,32], ochratoxin A (OTA) [38], fumonisins (FB_s_) [39], and patulin [23,38]. This is a first report that investigated the mycotoxin reduction ability of plant-derived LAB, since it has been involved in several traditional fermented dishes, especially in Asia and Southeast-Asia. Furthermore, it has been reported that it was also useful for immune modulation, the improvement of liver function, and the reduction in obesity [40]. In this study, many kinds of plant-derived LAB isolated from Thai fermented foods (33 strains) were assessed for ZEA removal under in vitro conditions. The results indicated that 17 strains of plant-derived LAB (approximately 10^9^ cfu/mL) were able to remove ZEA from liquid medium (0.5–23%). *L. plantarum* BCC 47723 isolated from wild spider flower pickle (Pag-sian-dorng) exhibited the highest ZEA removal (Figure 1). This suggested that ZEA removal by LAB was genus- and species-dependent. *L. plantarum* is a heterogeneous and versatile species encountered in a variety of environmental niches, including fermented food products, such as dairy, meat, fish, and vegetables, as well as plant matter. This species exhibits various biological effects such as antitumor, anticoagulant, antiviral, immune modulatory and anti-inflammatory, antidiabetic, and antioxidant or free radical scavenging activity. Long et al. [32] indicated that among *Lactobacillus* species isolated from rumen, the capability of ZEA removal by each bacterial cell (10^10^ cfu/mL of bacterial cells) was significantly different in the range of 26–69%. The highest ZEA removal was observed by *Lactobacillus mucosae*, followed by *Lactobacillus curvatus*, *Lactobacillus casei*, *Lactobacillus brevis*, and *Lactobacillus coryniformis*. This supported the idea that ZEA removal by LAB is species-dependent. The cell wall of general LAB is a complex assemblage of glycopolymers and proteins. It consists of a thick peptidoglycan that surrounds the cytoplasmic membrane and is decorated with proteins, polysaccharides, and teichoic acids [41]. The difference of sugars and amino acids in the glycopolymer or protein structures affected the pattern and/or structure of the bacterial cell surface in each one [41,42]. This means that the difference of the structure and components of the LAB cell wall have an important role in the capability of ZEA removal by LAB.

The physical and chemical treatments were used to investigate the possible components which are involved in the ZEA removal of plant-derived LAB. The role of proteins, polysaccharides, teichoic acids, and lipids on the bacterial cell wall of *L. plantarum* BCC 47723 was investigated through heat, chemical, and enzymatic treatments. SDS, urea, and pronase E were used to study the role of cell wall proteins on ZEA removal (Figure 2). The results indicated that ZEA removal capability was significantly increased after treatment with SDS, whereas in urea and pronase E-treated cells, this was not observed. This suggests that proteins on the bacterial cell wall were effective in ZEA removal by *L. plantarum* BCC 47723. The alteration of protein positions or amino acids was involved in ZEA removal. SDS denatures and breaks down proteins on the bacterial cell wall, exposing new ZEA binding sites. In contrast, El-Nezami et al. [31] reported that cell wall proteins had negligible involvement in ZEA binding by the *L. rhamnosus* strain GG (viable cell). The difference of these results confirms that ZEA removal by LAB depended on the type and structure of proteins on bacterial cell surface. M-periodate was selected to investigate the role of polysaccharides on ZEA removal since it can oxidize polysaccharides at the cis-OH position to aldehyde and acid [35]. The result indicated that no significant reduction was observed by m-periodate-treated cells when compared with an untreated control (Figure 2). This suggests that polysaccharides located on the bacterial cell surface did not affect ZEA removal and hydrogen bonds may be not involved in the interaction between *L. plantarum* BCC 47723 and ZEA. For the role of lipids on ZEA removal, the results revealed that the reduction in ZEA significantly decreased after bacterial cells were treated with lipase. Lipase will hydrolyze the ester bond lipid, resulting in a change of lipid structure. Similarly, Hernandez-Mendoza et al. [43] investigated the role of the bacterial cell membrane on mycotoxin adsorption. The results indicated that the protoplast (a bacterial cell with only a cell membrane without a cell wall) of *Lactobacillus reuteri* and *Lactobacillus casei* Shirota showed the capacity to reduce AFB_1_ in vitro. This study suggests that lipids were one of the components which were able to attach with the mycotoxin. In contrast, there were reports suggesting that the lipid elements on the bacterial cell surface of *L. rhamnosus* strain GG were not involved in ZEA [31] or AFB_1_ binding [35]. These research suggested that the effect of lipids on mycotoxin removal depended on the species of bacterial cell and types of mycotoxin. The diversity of the composition or elements on the bacterial cell wall structure may influence the capability of removing ZEA by LAB. According to the result mentioned above, it can be concluded that the bacterial cell surface were the elements responsible for the binding of mycotoxins by LAB [24,31,35,36,43].

*L. plantarum* BCC 47723 was also modified by heat treatment. The result demonstrated that the efficiency of ZEA removal was significantly increased (*p* < 0.05) after cells were inactivated (Figure 2). This result was consistent with the previous studies. Long et al. [32] found that the removal capacity of ZEA by heat-inactivated cell of *Lactobacillus mucosae* lm4208 was significantly increased, when compared with a control (viable cells). Similarly, the results from El-Nezami et al. [24] showed that heat treatment significantly enhanced the ability of *L. rhamnosus* strain GG and *L. rhamnosus* strain LC705 to remove ZEA from liquid medium. They found that no degradation products of ZEA were observed by high performance liquid chromatography (HPLC). This result suggests that binding could be the main mechanism for ZEA removal by heat-inactivated cells, not bio-transformation.

Polymyxin B is a polycation reagent, which is able to bind with ions on proteins and/or teichoic acids (anion polymers) in the structure of the bacterial cell wall surface. Therefore, it was chosen to investigate the role of electrostatic interaction in ZEA removal by *L. plantarum* BCC 47723. The results indicated that polymyxin B did not affect the capability of *L. plantarum* BCC 47723 to remove ZEA from liquid medium. This suggests that decrease in anions on the bacterial cell surface was not influenced by ZEA removal. Structurally, ZEA is a hydrophobic molecule and is estimated at pKa = 7.62. It is mainly in neutral form at pH 3.0 and the phenolate anion form is presented in solution at pH 8.0 [44]. Therefore, binding between bacterial cells and ZEA at pH 7.2 are not dependent on the alteration of ionization on bacterial cell surface. Haskard et al. [35] also supported that ionic interaction was not the main mechanism for AFB_1_ adsorption, since no change in AFB_1_ adsorption by *L. rhamnosus* strain GG was observed after the cell interacted with mono- and divalent ions. According to zeta potential values analysis, it was also confirmed that electrostatic interaction does not play an important role in ZEA removal by *L. plantarum* BCC 47723 (Figure 5). In contrast, there are many reports that have indicated that hydrophobic interaction plays an important role in the adsorption of mycotoxins, including AFB_1_ [35], patulin [45], and ZEA [31] by LAB. This is in accordance with our results for hydrophobicity (H_0_). Hydrophobic interaction was shown to involve the mechanism of *L. plantarum* BCC 47723 binding with ZEA in liquid medium (Figure 6). This result is also consistent with the data from Pearson’s correlation.

In order to investigate the alteration of the morphologies and element compositions on the bacterial cell surface after being physically and chemically treated, SEM-EDS was used in this experiment. The result suggests that the shape of bacterial cells in all treatments was obviously damaged and changed when compared with the untreated control (Figure 3). These alterations were related to ZEA removal by bacterial cells. This suggests that the chemical structures on the bacterial cell surface may be transformed to other structures, resulting in the efficiency of ZEA removal. The changes of the ratio of chemical elements on the cell surface were also observed after being treated with heat, chemicals, and enzymes. The results indicated that being treated with these strategies did not influence the changing of the chemical element ratio on the bacterial cell surface (Table 1).

The possible functional groups and the reaction which is involved in the ZEA removal process of bacterial cells was investigated using FTIR, zeta potential, and surface hydrophobicity (H_0_). Regarding the FTIR spectra, the results showed that the peak vibration patterns of each sample were basically the same either before or after ZEA adsorption (Figure 4), similar to the general pattern of LAB [23,46]. This result was consistent with Ge et al. [46], who indicated that the FTIR spectrum of *Lactobacillus brevis* 20023 was not changed after tenuazonic acid (TeA) adsorption. Apparently, bacteria did not completely lose their original structure after being adsorbed ZEA and the hydrogen bond was not involved in the ZEA removal. Nevertheless, the peak vibration of cells subjected to heat treatment was changed after being loaded with ZEA, in the range of 1250–850 cm^−1^. This region was dominated by C-OH, C-C, and C-O-C vibrations of polysaccharides and single form bending vibrations of the bonds in groups CH_2_ and CH_3_ present in teichoic acids, peptidoglycan, lipopolysaccharides, and phospholipids [47]. This suggests that ZEA removal may be involved in the polysaccharide groups in heat treatment which are located on the bacterial cell wall. This hypothesis was supported by Wang et al. [23], who indicated that patulin adsorption by heat-inactivated cells was predominantly related to carbohydrate components in the bacterial cell wall.

To assess the shelf-life of heat-inactivated cells on ZEA removal, freeze-drying was applied in this study since it has been a common technique used for the preservation of the microbial cells [48]. In this research, the freeze-dried heat-inactivated cells were collected in screw-cap microcentrifuge tubes and stored in a desiccator at room temperature (approximately 25 °C). After 90 days of storage, the capacity of ZEA removal by *L. plantarum* BCC 47723 was still active and stable (Figure 7). This suggests that the bacterial cell wall of heat-inactivated cells may not be destroyed, resulting in the percentage of ZEA removal not being changed.

## 4. Conclusions

Plant-derived LAB (*L. plantarum* BCC 47723) isolated from Thai fermented vegetable product had the capacity to reduce ZEA from liquid medium. *L. plantarum* BCC 47723 showed the highest removal of ZEA. Adsorption is a mechanism involved in ZEA reduction by plant-derived *L. plantarum* BCC 47723 since no degradation products such as its derivatives were observed by UHPLC analysis. The ZEA removal capability was species-dependent and depended on the type and component of proteins and lipids inside the bacterial cell wall structure, whereas no impact of polysaccharides was proven in this work. The FTIR spectra showed that the original structure of *L. plantarum* BCC 47723 was not completely lost after reacting with ZEA. Proteins were the main elements in the removal of ZEA. In addition, lipid structures on the bacterial cell surface were also shown to have the potential to adsorb ZEA. The interaction involved in ZEA reduction in *L. plantarum* BCC 47723 was hydrophobicity. These findings suggest that LAB derived from Thai fermented food was showed as a potential ZEA organic adsorbent to remove ZEA in foodstuff and animal feed. The capacity to reduce the ZEA on bacterial cells was protected by the freeze-drying technique. However, the assessment of ZEA desorption and ZEA removal by *L. plantarum* BCC 47723 in the digestion model and the identification of exact structures and the composition of the cell wall, which are responsible for ZEA adsorption, are required. The selection of species and strains that are additionally capable of inactivating mycotoxin will be further investigated.

## 5. Materials and Methods

### 5.1. Bacterial Strains and Chemical Reagents

Thirty-three LAB strains isolated from Thai fermented vegetables were used throughout this study. These strains, twenty-seven *Lactobacillus* spp., two *Pediococcus* spp., two *Enterococcus* spp. and two *Weissella* strains, were obtained from the Culture Collection of the National Center for Genetic Engineering and Biotechnology (BIOTEC), National Science and Technology Development Agency (NSTDA), Thailand. All strains were stored at −80 °C in 20% glycerol.

De Man Rogosa and Sharpe (MRS) medium (Merck, Darmstadt, Germany) was used to culture the 33 strains of plant-derived LAB. Chemical reagents including sodium chloride (NaCl), potassium chloride (KCl), sodium phosphate (Na_2_HPO_4_), and potassium phosphate monobasic (KH_2_PO_4_) were purchased from Carlo Erba Reagents (Bangkok, Thailand). Methanol (HPLC grade) was purchased from RCI Labscan (Bangkok, Thailand). Sodium dodecyl sulfate was purchased from Bio-Rad (Bangkok, Thailand). All enzymes and other chemical reagents were purchased from Sigma-Aldrich (St. Louis, MO, USA).

A solid standard of ZEA (5 mg) was purchased from Romer Labs (Singapore). It was dissolved in absolute methanol. A working solution was prepared to 2 µg/mL and stored in the dark at −20 °C in screw cap bottles until use.

### 5.2. Cultivation of LAB Strains

Thirty-three plant-derived LAB strains from the frozen stock culture were initially streaked onto MRS agar, and incubated at 30 °C for 48 h. Then, a single colony from each strain was transferred in MRS broth. After 18 h of incubation, 1% inoculum of bacterial cells was transferred and incubated again under the same conditions (30 °C for 18 h without shaking) in MRS broth. The bacterial cell densities of these cultures were measured using spectrophotometric assay at a wavelength of 600 nm and the volume of bacterial cells was adjusted by 0.01 M phosphate buffer saline (PBS buffer, pH 7.2), in order to obtain a bacteria concentration of approximately 10^9^ cfu/mL. The spread plate technique was also applied for bacterial density enumeration and confirmation.

### 5.3. Screening of Plant-Derived LAB to Remove ZEA 

According to Adunphatcharaphon [49] with slight modification, one milliliter of the cell cultures was centrifuged at 10,000 rpm for 10 min at 4 °C. The pellets (approximately 10^9^ cfu/mL) were washed twice with PBS buffer (pH 7.2) and then added with 1 mL of PBS buffer containing 0.2 µg/mL of ZEA standard working solution. After incubation at 30 °C for 1 h, the cell suspensions were centrifuged again and 100 µL of supernatant was analyzed with ZEA residues using an Enzyme-Linked Immunosorbent Assay (ELISA) (Romer Labs, Singapore, Singapore) at a wavelength of 450 nm.

### 5.4. Effect of Physical and Chemical Treatments on ZEA Removal 

To investigate the possible components of the bacterial cell which were involved in the ZEA reduction, heat treatment, chemical and enzymatic reagents (shown in Table 2) were used in this study. The method was performed followed by El-Nezani et al. [24] with slight modification. Briefly, *L. plantarum* BCC 47723 which exhibited the highest ZEA removal was cultured in the MRS broth at 30 °C for 18 h. After incubation, the cell suspensions were centrifuged at 10,000 rpm for 10 min at 4 °C. The pellets were washed twice with PBS buffer and then heated at 62 °C for 30 min or reacting with chemical and enzymatic reagents. The target of each reagent on the bacterial cell surface and the incubation conditions are also shown in Table 3. After reaction, the cell suspensions were centrifuged and then washed with phosphate buffer (pH 7.2). Finally, the pellets were used to testing ZEA removal using the ZEA ELISA test kit. Both cells before and after adding with ZEA standard solution were further characterized to study the mechanism involved in ZEA removal by plant-derived LAB. 

### 5.5. Characterization of ZEA Removal Mechanism by L. plantarum BCC 47723 

To understand the ZEA removal mechanism by plant-derived bacterial cells, both ZEA-exposed and -unexposed bacterial cells in all treatments (heat, chemical, and enzymatic treatments) were freeze-dried and then characterized using scanning electron microscopy coupled with energy dispersive X-ray spectroscopy (SEM-EDS) (SU-5000, HITASHI, Tokyo, Japan), Fourier transform infrared spectroscopy (FTIR) (Nicolet 6700 FT-IR Spectrometer, Thermo Scientific, Waltham, MA, USA), Zetasizer instruments (Nano ZS, Malvern, UK), and hydrophobic index (H_0_) measurement.

### 5.6. Study Shelf-Life of the Bacterial Cells on ZEA Removal 

To assess the shelf-life of bacterial cells as an organic adsorbent, heat-inactivated cells were collected by the freeze-drying technique using the FreeZone Plus 6 Liter Cascade Console Freeze Dry System (Labconco, Kansus City, MO, USA). Freeze-dried cells were stored at room temperature in a desiccator before testing for ZEA removal. The bacterial cells were used to check the ZEA binding activity after being kept for 0, 5, 10, 30, 60, and 90 days of storage.

### 5.7. Quantification of Zearalenone Using ELISA Test Kit 

All samples were diluted with methanol and buffer solution following the manufacturing procedure (Romer Labs, Singapore) before analysis of ZEA residues. PBS buffer solution containing 0.2 µg/mL of ZEA without bacterial cells was used as a control. The percentage of ZEA binding was calculated using the following equation:(1)$$\%\;\mathrm{ZEA}\;\mathrm{reduction}\;=\frac{\;\left[\mathrm{ZEA}\;\left(\mathrm{control}\right)-\;\mathrm{ZEA}\;\left(\mathrm{residues}\;\mathrm{in}\;\mathrm{sample}\right)\right]\times 100}{\mathrm{ZEA}\;\left(\mathrm{control}\right)}$$

### 5.8. Confirmation of ZEA Using UHPLC 

Ultra-high performance liquid chromatography (UHPLC) was used to confirm the ZEA removal by *L. plantarum* BCC 47723. ZEA analysis was performed following the method of Avantaggiato et al. [50] with slight modification. ZEA was analyzed using an Agilent 1290 UHPLC system equipped with a photodiode array (DAD) and a spectrofluorometric (FLR) detector. The analytical column was an Acquity UPLC^®^ BEH C18 (50 mm × 2.1 mm, 1.8 µm particle column preceded by an in-line filter (0.3 µm). An isocratic mobile phase comprising a mixture of water/methanol (85:15, *v/v*) and acetic acid (1%) was eluted at 0.35 mL/min for 8 min. The injection volume was 10 µL (full loop mode). The UV absorption spectrum of ZEA was recorded in the range of 190–400 nm. UV absorbance data were collected at wavelengths of 274 nm. ZEA retention time was 3.75 min. The ZEA concentration was calculated by the peak area.

### 5.9. Statistical Analysis 

All experiments were performed in triplicate. Statistical analysis was carried out using SPSS version 17.0 for Windows and significant difference (*p* < 0.05) between means were determined by Duncan’s multiple range test. The correlation between parameters in ZEA removal by each treatment was analyzed using Pearson’s correlation coefficient (PCC).

## Figures and Tables

**Figure 1 toxins-13-00286-f001:**
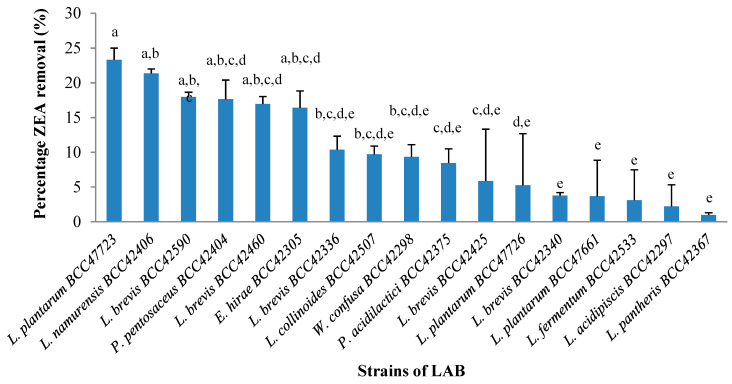
Percentage of ZEA removal from buffer solution by plant-derived LAB. The experiment was performed using a bacterial concentration approximately 10^9^ cfu/mL at 30 °C for 1 h in PBS buffer containing ZEA 0.2 µg/mL. Means (n = 3) with different letters are significantly different according to Duncan’s multiple range test (*p* < 0.05).

**Figure 2 toxins-13-00286-f002:**
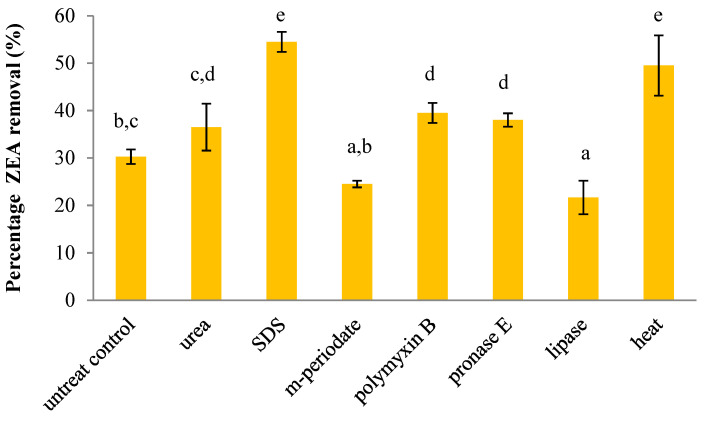
Effect of physical and chemical treatments on ZEA removal by *L. plantarum* BCC 47723. Values are means ± standard deviations of triplicate experiments. Different letters (a–e) indicated a significant difference at *p* < 0.05 in each physical and chemical treatments of bacterial cells.

**Figure 3 toxins-13-00286-f003:**
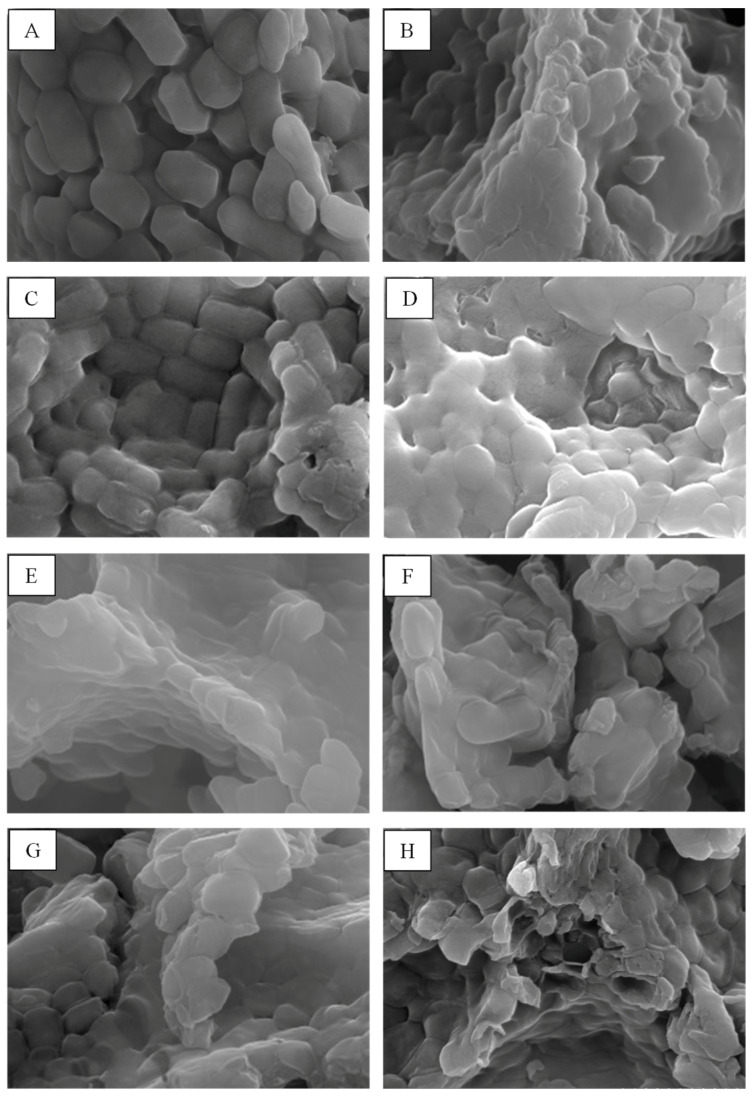
SEM photographs of bacterial cells at a magnification of 20,000×: (**A**) untreated of *L. plantarum* BCC 47723; (**B**) urea treatment; (**C**) SDS treatment; (**D**) m-periodate treatment; (**E**) polymyxin B; (**F**) pronase E treatment; (**G**) lipase treatment; (**H**) heat treatment.

**Figure 4 toxins-13-00286-f004:**
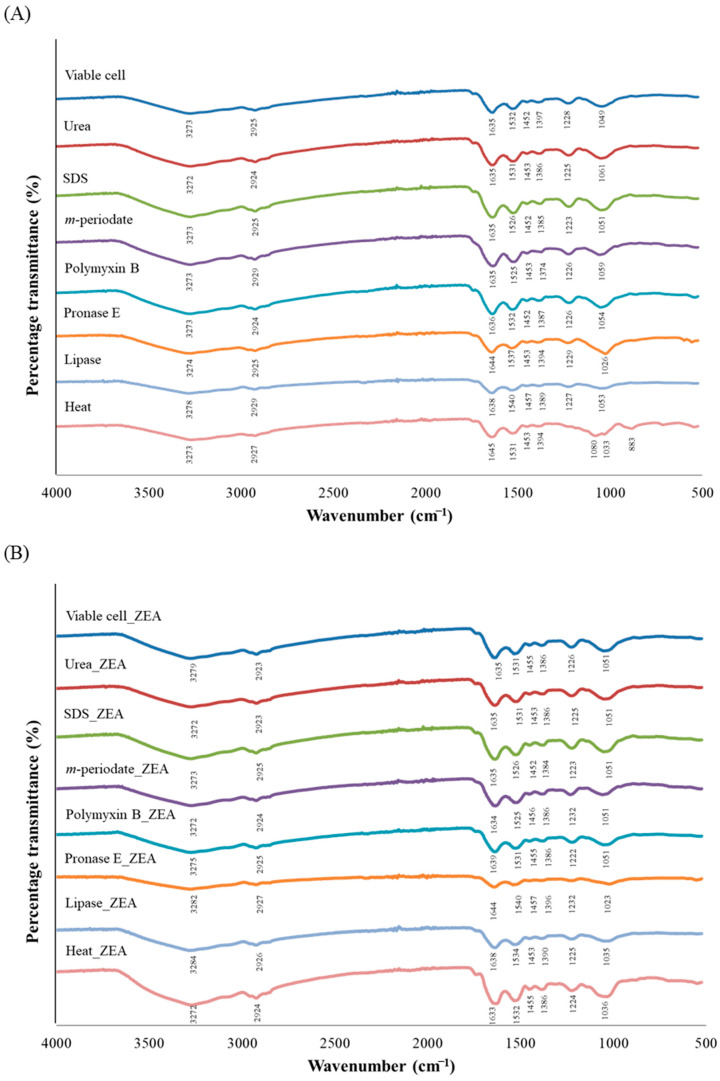
FTIR spectra of ZEA-unexposed (**A**) and ZEA-exposed bacterial cells (**B**) in each treatment.

**Figure 5 toxins-13-00286-f005:**
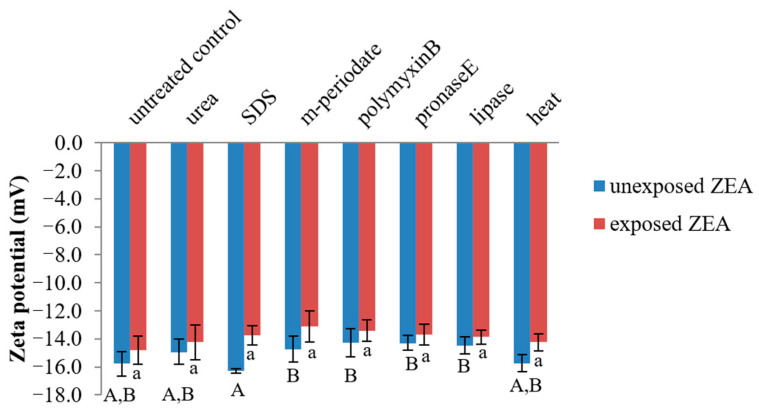
Zeta potential of ZEA-unexposed and ZEA-exposed bacterial cells in each treatment. The capital letter (A,B) represents a significant difference at *p* < 0.05 in each treatment of ZEA-unexposed bacterial cells and the letter (a) represents a significant difference at *p* < 0.05 in each treatment of ZEA-exposed bacterial cells.

**Figure 6 toxins-13-00286-f006:**
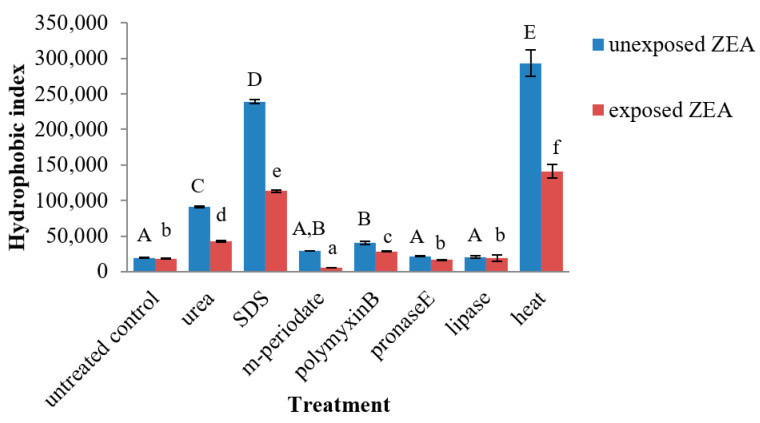
Surface hydrophobicity (H_0_) of bacterial cells in each treatment. Different capital letters (A–E) indicated a significant difference in treatment of ZEA-unexposed bacterial cells at *p* < 0.05. Different letters (a–f) indicated a significant difference in treatment of ZEA-exposed bacterial cells at *p* < 0.05.

**Figure 7 toxins-13-00286-f007:**
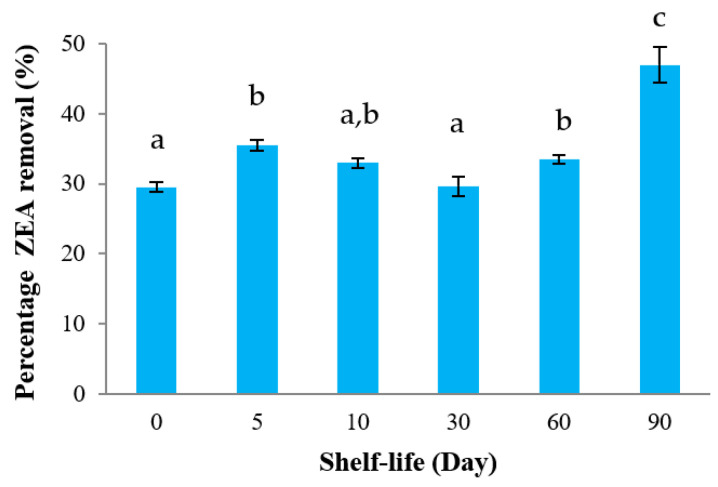
The shelf-life of heat-inactivated cells of *L. plantarum* BCC 47723 on ZEA removal. Values are means ± standard deviations of triplicate experiments. Different letters (a–c) indicated a significant difference at *p* < 0.05 in each day.

**Table 1 toxins-13-00286-t001:** The EDS analysis of bacterial cells after being treated with physical and chemical treatment.

Treatment	The Ratio of Each Chemical Elements											N/C Ratio
	C	O	N	P	K	Na	Cl	Al	S	I	Ca	
Control	56.90 ^a,b^	20.03 ^a,b^	12.30 ^a^	4.16 ^e^	2.63 ^e^	2.30 ^f^	1.40 ^c^	0.07	0.00 ^a^	0.00 ^a^	0.00 ^a^	0.22 ^a,b^
Urea	54.03 ^a^	24.03 ^c^	17.03 ^b,c^	2.47 ^c,d^	2.00 ^c,e^	0.30 ^a,b,c^	0.00 ^a^	0.03	0.00 ^a^	0.00 ^a^	0.00 ^a^	0.32 ^c^
SDS	66.63 ^c^	17.73 ^a^	13.26 ^a,b^	0.87 ^a^	1.23 ^b^	0.17 ^a^	0.00 ^a^	0.03	0.03 ^a,b^	0.00 ^a^	0.00 ^a^	0.20 ^a^
*m*-periodate	55.40 ^a,b^	21.20 ^a,b,c^	16.63 ^b,c^	1.93 ^b,c^	2.70 ^e^	0.40 ^b,c^	0.00 ^a^	0.00	0.20 ^d^	1.47 ^b^	0.00 ^a^	0.30 ^b,c^
Polymyxin B	55.13 ^a^	21.43 ^a,b,c^	17.97 ^c^	2.87 ^d^	1.80 ^b,c^	0.67 ^e^	0.00 ^a^	0.00	0.10 ^c^	0.00 ^a^	0.00 ^a^	0.33 ^c^
Pronase	62.03 ^b,c^	21.30 ^a,b,c^	11.97 ^a^	2.03 ^b,c^	1.90 ^c^	0.27 ^a,b^	0.00 ^a^	0.00	0.00 ^a^	0.00 ^a^	0.56 ^b^	0.19 ^a^
Lipase	58.07 ^b^	21.06 ^a,b,c^	14.07 ^a,b,c^	2.27 ^d^	3.30 ^d^	0.43 ^c^	0.00 ^a^	0.03	0.07 ^b,c^	0.00 ^a^	0.00 ^a^	0.24 ^a,b,c^
Heat	55.7 ^a,b^	21.73 ^b,c^	17.77 ^c^	1.37 ^a,b^	0.20 ^a^	2.27 ^f^	0.93 ^b^	0.00	0.00 ^a^	0.00 ^a^	0.00 ^a^	0.32 ^c^

^a–c^ Letters of each column indicated significant differences between treatments at *p* < 0.05. Values are means ± standard deviations of triplicate experiments.

**Table 2 toxins-13-00286-t002:** Pearson’s correlation coefficient between parameters related to ZEA removal (n = 3).

		ZEA Removal	C	O	N	P	K	Na	Cl	Al	S	I	Ca	N/C	H_o_	Zeta
ZEA removal	Pearson Correlation	1	0.644	−0.436	−0.094	−0.475	−0.967 **	−0.288	−0.211	−0.317	−0.392	−0.411	0.108	−0.226	0.834 *	−0.495
	n	7	7	7	7	7	7	7	7	7	7	7	7	7	7	7

* Correlation significant at the 0.05 level (2-tailed). ** Correlation significant at the 0.01 level (2-tailed).

**Table 3 toxins-13-00286-t003:** Incubation conditions of each physical, chemical, and enzymatic treatments.

Treatment	Target	Buffer	Incubation Conditions	
			Temp. (°C)	Time (h)
8 M urea	Proteins	Distilled water	37	1
0.1 M SDS	Proteins	Distilled water	37	1
*m*-periodate (10 mg/mL)	Polysaccharides	Distilled water	37	2
Polymyxin B (10 µg/mL)	Teichoic acids	Distilled water	37	4
Pronase E (0.5 mg/mL)	Proteins	Phosphate buffer	37	2
Lipase (0.5 mg/mL)	Lipids	Phosphate buffer	37	2
Heat	Bacterial cell wall	Phosphate buffer	62	0.5

## Data Availability

The data presented in this study are available on request from the corresponding author. The data are not publicly available due to privacy restrictions.

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
