# Peer review of "In Vitro Mechanism Assessment of Zearalenone Removal by Plant-Derived Lactobacillus plantarum BCC 47723"

_toxins, 2021, doi:10.3390/toxins13040286_

Round 1

Reviewer 1 Report

In the current work the authors investigated the efficacy of the mycotoxin, zearalenone (ZEA), removal by the different strains of Lactobacillus plantarum. The authors also demonstrate possible mechanisms by which L. plantarum mediates ZEA. Overall, the study has merits, but the authors should significantly improve the quality of writing and data presentation. Please see my specific comments below.

Line 4: add ‘a’ after the words, ‘Zearalenone is’

Line 7: put a ‘,’ after the words, ‘fertility problems’

Line 15-16: the authors should specify whether the effect was a positive/negative/neutral: ‘proteins and lipids…..obviously affected’

The abstract started well but ended kind of abrupt. May be a concluding sentence that summarizes the overall finding will improve the quality of the abstract.

Add a sentence at the end of the introduction to give an overall perspective of the interesting results obtained from this study.

Results:

Line 70: replace ‘according to’ with ‘Out of’

Line 72-77: italicize the genus and species names of Lactobacillus sp. and also through out the text and in the related Figs.

The genus and species names should be italicized in fig. 1.

Line 80: the authors should specify N= ? (biological or technical replicates) in the fig. legend.

Line 83: the authors should include a sentence in the beginning on why these different treatments were used to estimate the percentage of zea removal. Add the word ‘being’ between ‘after’ and ‘treated’

Line 92: Fig. 2 legend, how many replicates used?

Line 94: the authors should mention at the beginning of the paragraph on why these different treatments were used for this part of the study.

Line 104: ‘could affected’: use proper tense

Line 105: any abbreviation used in the text e.g. EDS, should be spelled out first prior to the use of the abbreviation. I see it is explained in the ‘Materials and Methods’, but this section comes later in the text. Is there any data that support the statement, ‘EDS analysis showed……O, N, P, K, and Na’?

Line 118: replace ‘excepting’ with ‘except’

Line 158: The authors should mention why it was important to measure the shelf-life of bacterial cells upon ZEA removal. For both Table 1 and Fig. 7 what are the values of N? (replicates?).

Line 163: Fig. 7 shows that the percentage of ZEA removal varies with shelf-life but no significant differences were shown in this Fig. Also, N? (replicates?).

Discussion:

Throughout the discussion whenever any result is discussed, it should be specified by appropriate Fig. or Table nos. This aspect is missing in the entire discussion section.

Conclusions:

This can be shortened with more focused statements. The authors should also mention the utility of this work in solving real life problems.

Line 394: ‘all experiments were’: correct statement.

Materials and methods:

Appropriate citations should be provided throughout this section.

Reviewer 2 Report

This study evaluate the ZEA removal capacity of plant-derived lactic acid bacteria (LAB) and to investigate the possible components and mechanisms involved in the removal of ZEA by physically and chemically treated plant-derived LAB. The introduction is relevant, but it's very short, not enought information on previous studies.

The methods are appropriate and the results are clear.

Why did you decide to use only one pH for testing? Have you thought about testing desorption? in the conclusions you have not hypotized whether the zea remains as it is or is degraded.

Reviewer 3 Report

The manuscript presents an interesting, practical and useful study that demonstrates that lactic acid bacteria of the genus Lactobacillus can be used to inactivate mycotoxins, such as zearalenone produced by fungi. The authors explain the reduction of zearalenone by the specific action of proteins and lipids on bacterial cell structure.

In the Introduction the authors describe the main ways of reduction of zearalenone by microorganisms, through adsorption, the action of enzymes or the modification of the molecular structure of this harmful mycotoxin. The purpose of the research is mentioned.

The results describe the screening of plant-derived lactic bacteria, effect of physical and chemical treatments, and characterization of ZEA removal mechanism.

Observations (in order of the text)

  1. All microbial species should be written in italics (for example: lines 71-78, 100, 186, 219, figure 1 - Lactobacillus plantarum, Lactobacillus mucosae, Lactobacillus curvatus, Lactobacillus casei, Lactobacillus brevis, and Lactobacillus coryniform and Lactobacillus casei)

  1. For the results shown in fig. 1 it would be good to specify the density of the bacterial cells tested, even if the procedure is described in Materials and Methods. If removal of ZEA is due to the phenomenon of adsorption on cellular coatings, there should be a direct relationship between the number of cells and the percentage of ZEA removal. It should be specified whether such studies have been made or exist in the literature. Also, in the studies carried out in the following period it should be observed if, in addition to the removal of ZEA by adsorption, microbial enzymes are also involved.

  1. In Fig. 4 the percentage units should be written on the Oy axis.

  1. It is mentioned that the determination of zealarenone was made after 1 hour of incubation at 30 ° It should be explained why this time of action of bacteria on mycotoxin was chosen and if tests were performed for other incubation times. Also, in my opinion, in the Results section, this action time should still be mentioned, even if the method is described in the Materials and Methods section.

  1. 7 - it should be specified if the authors have any explanation for the increase in the percentage of ZEA removal after 90 days of bacteria storage.

  1. It would have been good for the number of bibliographic references in the recent period to be higher, those from 2015-2021 represent less than a quarter.

Given that L. plantarum strains are used to preserve fodder, the reduction of ZEA (based on adsorption phenomena at the cellular wall) should be done by removing bacteria from these feeds, a rather difficult operation. Therefore, the studies should be continued for the selection of species and strains that are additionally capable of inactivating mycotoxin.

Reviewer 4 Report

Dear Authors, Dear Editor,

I was invited to review the manuscript «In vitro Mechanism Assessment of Zearalenone Removal by 2 Plant-derived Lactobacillus plantarum BCC 47723».

In their study, the authors evaluated removal ability of Lactobacillus plantarum BCC 47723 against Zearalenone (ZEA) that is harmful secondary fungal metabolite, produced by fungi mostly belonging to the genus Fusarium. The main objectives of the authors were to evaluate the ZEA removal capacity of plant-derived lactic acid bacteria (LAB) and to investigate the possible components and mechanisms involved. Authors show that 17 out of 33 plant-derived LAB exhibited ZEA removal from liquid medium with percentage of removal ranging from 0.5-23% and with Lactobacillus plantarum BCC 47723 exhibiting the highest removal. According to data proteins and lipids on bacterial cell structure were responsible of ZEA removal by L. plantarum BCC 47723.

Here are my comments :

Major : As indicated by the authors, the removal ability of L. plantarum on ZEA is already known and described. So, I don’t see how and why this study is original or brings new informations. Please justify and indicate what is new here.

Minor :

  • Name of bacteria has to be indicated in italic.

  • Figure 2 : it is unclear if the % indicated in Figure 2 are % of removal : are treated-bacteria showing in higher removal capacity than live bacteria ? Or the % indicated in Figure 2 are % of the 23 % removal capacity of the strain ?

Regards

Round 2

Reviewer 4 Report

Dear Authors, Dear Editor,

Thanks to the authors that addressed all my comments. 

Regards